# Subjective Well-Being, Health and Socio-Demographic Factors Related to COVID-19 Vaccination: A Repeated Cross-Sectional Sample Survey Study from 2021–2022 in Urban Pakistan

**DOI:** 10.3390/ijerph20166545

**Published:** 2023-08-08

**Authors:** Khadija Shams, Alexander Kadow

**Affiliations:** 1Department of Economics, Shaheed Benazir Bhutto Women University, Peshawar 25000, Pakistan; 2Department of Economics and Law, Frankfurt University of Applied Sciences, 60318 Frankfurt am Main, Germany; kadow@fb3.fra-uas.de

**Keywords:** COVID-19 vaccine, quality of life, aging, epidemiology, health status, subjective well-being, socio-demographic factors, urban Pakistan

## Abstract

Background: Containing the spread of the COVID-19 rests on many people willing to get vaccinated. At the same time, it is important to recognize the various socio-demographic factors associated with COVID-19 vaccination. This paper aims to identify socio-demographic and health factors related to the COVID-19 vaccine and its impact on subjective well-being in urban Pakistan. Methods: Pooled cross-sectional sample surveys collected in 2021 and 2022 (*n* = 4500 households) via a questionnaire provided to household’s heads. In each wave, data were collected using the same methodology, sample size and sampling techniques (proportional stratified random sampling). An ordered probit regression model was used to identify the various socio-demographic and health factors related to the COVID-19 vaccine and its impact on subjective well-being. Sample weights were applied to all the regression analyses to improve population generalizability. Results and conclusion: Besides socio-demographic factors such as being healthy, educated and richer, coronavirus vaccination plays a positive and significant role in overall subjective well-being. However, vaccination has a smaller effect on men or older populations compared to women or younger populations in terms of their subjective well-being. Moreover, as expected, the vaccination has the strongest positive effect on the healthy population and its subjective well-being.

## 1. Introduction


**What Is Already Known on This Topic**


The existing research on the issue states that women and younger people suffered more from the pandemic compared to men and older people, respectively. This study addresses issues within the framework of a life satisfaction/happiness model with particular reference to urban Pakistan.


**What This Study Adds**


We found that men and older people suffered more from the pandemic compared to women and younger people in this part of the developing world.


**How This Study Might Affect Research Practice or Policy**


The results can help direct the Pakistani government’s efforts in dealing with psychosocial and health problems stemming from the COVID-19 pandemic, especially for men and older people.


**Country Context**


Pakistan’s official name is the Islamic Republic of Pakistan. It is the world’s 33rd biggest country, covering an area of 796,095 square kilometers (307,374 square miles). It shares a 1046 km (650-mile)-long coastline with the Arabian Sea and a sea border with Oman. It shares borders with India, Afghanistan, Iran and China in the east, west, south-west and north-east, respectively. To the north-west, Afghanistan’s Wakhan Corridor just separates it from Tajikistan. It stands as the world’s fifth most populous country. It has a population of more than 242 million. The annual population growth rate is 1.91%. It has an urban population accounting for 38% of the total population. The estimated annual growth rate of urbanization from 2020 till 2025 is 2.1%. It holds the second-biggest Muslim population in the world. The sex ratio at birth is 1.05 male(s)/females; 48.7% of Pakistan’s population is female, while 51.3 percent of the population is male. The percentage of the population that is elderly (65 years and over) is 4.77%. The life expectancy at birth is 67.89 years for males and 72.14 years for females. The total fertility rate is 3.39 children born/woman. The overall median age is 22 years: for males, 21.9 years, and for females, 22.1 years. The total youth unemployment rate (ages 15–24) is 9.4%: 9.7% for males and 8.2% for females. The total literacy rate is 58%: 69.3% for males and 46.5% for females [1]. The average monthly household income is PKR 41,545 [2]. The adult population (18 years and above of age) of Pakistan is 59.7%, approximately, and 59% are fully vaccinated [3].

The outbreak of the novel coronavirus has brought considerable human suffering and major economic disruption. At this stage, it is hard to imagine what further damage the virus may cause to the global economy as many countries are still trying to control the ongoing pandemic through vaccination, strict social distancing rules and other safety measures. In particular, developing countries have suffered a lot, and there are already fears that income and health inequalities among the rich and the poor may widen in the long run. This gives increasing relevance and scope to the economics of happiness. Happiness is the subjective sense of well-being, which is based on self-reported measures of so-called subjective well-being (SWB). Using self-reports to measure subjective well-being is standard in psychology but less common in economics. In the remainder of this paper, we shall use SWB, subjective life satisfaction and happiness interchangeably. The concept of happiness or perceived life satisfaction involves health and socio-demographic analysis that includes various aspects like age, gender, income, health, marital status, religion, education, social capital, political and social institutions and services, social and interpersonal relationships, diet, other personal characteristics, regional characteristics and country characteristics [4,5,6,7,8]. The basic findings for advanced economies from the United States and European panel datasets state that SWB increases with increases in education, income, health, sexual activity, religious involvement, being female, living as a married couple, having no children and being married to one person at a time, i.e., monogamy. Moreover, happiness observes a U-shaped behavior with the increase in age, with a turning point in the mid- to late 40s. On the other hand, SWB decreases with being unemployed, sexually inactive, having children, living in separation or being divorced, being less educated, having poor health, having lower income and living as immigrants, minorities or commuters [9,10]. Happiness captures all domains of life [11,12,13]. It is considered that income and wealth ensure higher education and improve health status, which positively affects life satisfaction or SWB [12]. Other things being equal (ceteris paribus), higher income ensures a higher level of happiness but at a diminishing rate [6,7,8,9,10,11,12,13,14,15]. Moreover, COVID-19-vaccinated people surveyed in Tokyo, New York and Shanghai report better social and psychological well-being and expect better earnings than the unvaccinated [16]. The present study will try to explore the above-mentioned insights for a developing economy like Pakistan with a particular view on the impact of COVID-19 vaccinations. Multiple studies have shown the significance of socio-demographic characteristics, e.g., women and younger people have a lower acceptance rate of the COVID-19 vaccine [17] and those for other diseases in the past. We confine our analysis in this paper to the urban areas of this major emerging economy, which have also been hit hard by the virus. The aim of this study is to evaluate the effect of COVID-19 vaccination and the associated socio-demographic factors on the subjective well-being of households in urban Pakistan.

## 2. Methods

### 2.1. Study Design and Setting

The pooled dataset used in the analysis is based on two datasets for urban Pakistan using the sample household surveys conducted in 2021 and 2022 between March and May for each year. Using proportional stratified random sampling methods, our dataset includes all of the four provinces of Pakistan: Punjab, Sind, Baluchistan and Khyber Pukhtunkhuwa (KP). The households were selected within those strata using proportional random sampling techniques. In both waves, we applied similar selection criteria, sample size, research methodology and sampling techniques for data collection. The sample includes households from eight major cities that comprise approximately two-thirds of the total number of major cities across the country. A sample size of 2250 households per period (i.e., Survey 2021, *n*_1_ = 2250, and Survey 2022, *n*_2_ = 2250) was attained, leading to a pooled sample size *n* = 4500 households (*n* = *n*_1_ *+ n*_2_). Using a proportional allocation of the sample based on the population figures of the provinces and the sampled cities [18], 2250 households (50% of urban population in each wave) from Punjab: 1125 in Lahore, 450 in Faisalabad, 338 in Rawalpindi, 225 in Multan and 112 in Islamabad; from Sind, 1800 households in Karachi (40% of urban population in each wave); from KP, 270 households in Peshawar (6% of urban population in each wave) and from Baluchistan, 180 households in Quetta (4% of urban population in each wave) were included in the pooled sample. Furthermore, in order to ensure a good representation of urban Pakistan, sample weights (pweights) were given to the sample households corresponding to their cities. The weighting scheme is provided in Appendix A. The same sample weights were applied to all the regression analyses of the study. Moreover, the sample weights are based on the latest available 2017 census of Pakistan [18]. 

The data were collected through questionnaire and informed consent from potential research participants was obtained. All questions correspond to the household’s head (i.e., an adult person or member of the household, male or female, age 18 years or above) that is responsible for making general decisions, organization and care of the household. Based on [14,15,16,17,18,19], all the questions are valid or relevant to the present study and the data were cleaned to avoid any duplication in the data sets. Moreover, the happiness measure used in this study might have an overlap with satisfaction with life scale (SWLS), which is the most widely used scale and contains five items and also has good validity and reliability [20]. 

An Excel sheet with the data collected was generated. Data were entered and filtered as per inclusion criteria. Then, the Excel sheet was shifted to statistical software package STATA 17 for statistical analysis. The summarize command was used to calculate descriptives (mean, percentages or frequencies and standard deviation). Ordered probit regression analysis was used to estimate the effect of vaccination and the associated socio-demographic factors on subjective well-being.

### 2.2. Outcome Variable and Covariates

In order to assess self-perceived subjective well-being, or SWB, we need to construct a happiness metric. Consistent with the existing happiness literature [14,19,21], we construct an ordinal scale-based happiness metric such that a higher index accounts for higher SWB and vice versa. We do this by asking the head of household the following question: “*What is your level of happiness from your existing life as a whole?*” Responses were reported on a scale from 1 to 4, with 1 representing “*Not at all happy*”*;* 2 points for “*less than happy*”; 3 points for “*Rather Happy*” and 4 points for “*Extremely Happy*”.

Table 1 displays descriptive statistics of the happiness index. We observe that seventy percent of the observations are placed at the lower ends (i.e., “*Not at all happy*” and “*Less than happy*”) of the happiness scale, which are approximately equally distributed. Similarly, the upper ends of the happiness scale (i.e., “*Extremely happy*” and “*Rather happy*”) are roughly equally dispersed and hold thirty percent of the data, as shown in Table 1. 

Table 2 provides descriptive statistics of the sample. We observe almost sixty percent representation of male and forty percent female respondents in the given sample. Forty-nine percent of the sample lives in married couples. Average age and education is thirty-four and twelve years, respectively. Forty-six percent of the respondents self-assessed as being healthy and sixty-two percent reported themselves to be vaccinated (including partially and fully vaccinated, both). The childless and unemployed households account for forty-two and forty-five percent of the data, respectively. The average absolute nominal monthly household income is two hundred and sixty-four US dollars (USD). 

### 2.3. Statistical Analyses

In order to analyze the subjective well-being during the ongoing pandemic in the given time period (2021, 2022), we resort to the well-established econometric model in the literature of happiness studies. For instance, according to Bruni and Porta [19] and White, Gaines and Jha [22], happiness or SWB depends on different socio-economic and demographic factors. In the light of the given literature, we apply the baseline Model (1) to analyze SWB for the given study area as follows:(1)$$\begin{array}{c} {SWB}_{i}=\beta_{0}+\beta_{1}{\left(sex_{i}\right)}_{dummy}+\beta_{2}\left(age_{i}\right)+\beta_{3}{\left(age_{i\;}\right)}^{2}+\beta_{4}\left(education_{i}\right)\\ \;\;\;\;\;+\beta_{5}{\left(unemployed_{i}\right)}_{dummy}+\beta_{6}ln\;\left(monthly\;income_{i}\right)+\beta_{7}{\left(childless_{i}\right)}_{dummy}\\ \;\;\;\;\;+\beta_{8}{\left(married\;couple_{i}\right)}_{dummy}+\;\;\beta_{9}{\left({healthy}_{i}\right)}_{dummy}\\ \;\;\;\;\;+\beta_{10}{\left(married\;couple_{i}.children_{i}\right)}_{dummy}+\beta_{11}{\left({vaccinated}_{i}\right)}_{dummy}\\ \;\;\;\;\;+\beta_{12}{\left(region_{i}\right)}_{dummy}+\beta_{13}{\left({sex}_{i}\;.\;{vaccinated}_{i}\;\right)}_{dummy}+\beta_{14}{\left({age}_{i}\;.\;{vaccinated}_{i}\right)}_{dummy}\\ \;\;\;\;\;+\beta_{15}{\left({income}_{i}\;.\;{vaccinated}_{i}\;\right)}_{dummy}+\beta_{16}{\left({married\;couple}_{i}\;.{vaccinated}_{i}\;\right)}_{dummy}\\ \;\;\;\;\;+\beta_{17}{\left({healthy}_{i}\;.{vaccinated}_{i}\;\right)}_{dummy}+\beta_{18}{\left({unemployed}_{i}\;.\;{vaccinated}_{i}\;\right)}_{dummy}\\ \;\;\;\;\;+\beta_{19}{\left({childless}_{i}\;.{vaccinated}_{i}\;\right)}_{dummy}+\beta_{20}{\left({education}_{i}\;.{vaccinated}_{i}\;\right)}_{dummy}+\;\varepsilon_{i}+\\ i=1,2,3\ldots 4500\;\mathrm{households} \end{array}$$

Our rich dataset allows different potential determinants of well-being. The data available to us permit distinguishing between different potential determinants of happiness. As our SWB metric follows an ordinal scale of measurement, we apply ordered probit regression analyses to estimate Model (1). The explanatory variables include: sex, age (in years), education (in years), employment status (considering that the respondents, if employed, can work from home in compliance with the social distancing and other safety measures by the government over the current study period), health status, marital status, vaccination status, number of children, household’s monthly income (given in Pakistani rupees and measured in natural logs) in pure, absolute and nominal terms and region of household *i*. In our model, we have several binary or dummy variables such as gender, unemployment, marital status and childlessness. These are coded as 1 if the respondent is male, unemployed, living as a married couple and childless, and 0 otherwise. Regional background corresponds to three mutually exclusive dummy variables for households living in Punjab, Sindh and KP. The reference category corresponds to households belonging to Baluchistan.

The health status was assessed by asking the head of the household the following question: “*How would you assess your current overall health status?*” Responses were recorded as “*healthy*” and “*unhealthy*” and were coded as 1 and 0, respectively. The corona vaccination status was asked with the following question: “*What is your current vaccination status against Corona virus?*” Answers were recorded as “*vaccinated*” and “*not vaccinated*” and were coded into 1 and 0, respectively. The interaction effect between health status and vaccination status is given by the coefficient *β*_17_ that captures the moderating effect of vaccination on the relationship between the respondent’s self-reported health status and his/her overall subjective well-being. Furthermore, the coefficients *β*_13_, *β*_14_, *β*_15_, *β*_16_, *β*_18_, *β*_19_ and *β*_20_ show the interaction effects between being vaccinated (coded as 1 for being vaccinated and zero otherwise) and the corresponding set of control variables, e.g., sex (coded as 1 for male and 0 otherwise); age (if older than 40 years (middle age) coded as 1 and 0 otherwise); income (coded as 1 if above than average monthly income of 264 US dollars and 0 otherwise); marital status (coded as 1 if living as a married couple and 0 otherwise); employment status (coded as 1 for being unemployment and 0 otherwise); being childless (coded as 1 for being childless and 0 otherwise) and education (coded as 1 if above average education of twelve years and 0 otherwise), respectively. 

Many happiness studies, both on developed and developing countries, incorporate children as one of the determinants of happiness, although with mixed evidence. For instance, Blanchflower [9] and Tella, MacCulloch and Oswald [23] find that with the increase in the number of children, household’s happiness decreases. On the other hand, Clark [24] and Clark, Frijters and Schields [25] find no effect, and Stutzer and Frey [26] report positive effects of children on the household’s happiness. Frey and Stutzer [15], based on Swiss household survey data, observe that children have a negative impact on the happiness of single parents, while having children barely affects the happiness of married couples. This evidence, however, refers to developed countries. We pursue it further with regard to the COVID-19 pandemic for the study area over the given period (2021, 2022) using Model (1). More specifically, the coefficient *β*_10_ captures the interaction effect of living as married couple and having children on the respondent’s happiness, such that the interaction between the two variables is coded as 1 if the respondent is living in a married couple and has children in the household or 0 otherwise. Note that children are considered here as household members who are less than 16 years old.

## 3. Results and Discussion

We apply ordered probit regressions to estimate the baseline Model (1) without and with vaccination effect, using the statistical software STATA 17. The results are given in Table 3. As expected, our results indicate that SWB is higher among richer, healthier and more educated individuals. Similarly, we find that those who are vaccinated against COVID-19 report higher SWB compared to those who are not vaccinated against the virus. In contrast, those who are unemployed report lower SWB compared to their counterparts. Our results corroborate those of Sen [27]; Guardiola and Garcia-Munoz [28]; Kingdon and Knight [29]; Knight, Song and Gunatilaka [30]; Rojas [31,32] and Pradhan and Ravallion [33], who suggest that health, enlightenment through education and certain livelihood parameters (e.g., living standard and size of land holdings, etc.) improve one’s capabilities to access public services, which in turn have a positive influence on self-reported life satisfaction or SWB. 

Children and marital status are variables for which it is less straightforward to develop prior expectations. Most of the literature on developed countries suggests that children have potential negative effects on SWB of the households [9]. One possible explanation could be the financial burden and extra parental responsibilities on the part of their parents. However, our results suggest a positive effect of having children in a household on the SWB of the respondents. Children in the developing world are usually considered as an insurance mechanism against the economic risks in case of less government support for old age and after retirement. As far as marital status is concerned, generally speaking, married couples tend to be happier compared to those who are single, divorced, separated, widow or widower [9,30,34]. Our results are in line with the existing literature and support the notion that living as a married couple increases the SWB of the household. The interaction effect between children and marital status is denoted by the coefficient *β*_10_. The interaction coefficient *β*_10_ also suggests that married couples living with their children are happier with their lives, particularly during the pandemic when living together in stable family structures seems to have positive effect on SWB [35,36]. According to the existing happiness literature, age effects are usually non-linear. For instance, Blanchflower [9], based on US and European panel data, suggests that happiness is U-shaped based on age with a turning point of mid- or late forty years of age, approximately. That is, happiness first falls sharply towards mid- or late age and then recovers positively towards retirement. In contrast, for our study period (2021, 2022) during the pandemic, we establish an inverted U-shaped curve of happiness with increasing age, with a statistically significant tipping point at forty years of age, approximately. 

### Corona Vaccination Effect

On a more general note, the COVID-19 vaccination has a positive effect on the SWB of the respondents as shown in Table 3. The ordered probit regression analysis of Model (1) with vaccination-interaction effects is given in Table 4. The interaction coefficient (*β*_13_) of being male and being vaccinated against the virus is found to be negative and statistically significant, which indicates that, compared to females, males were disproportionally affected in terms of SWB due to vaccination. One possible explanation could be that in most of the developing countries, men are usually the breadwinners and have more social interactions compared to women, which may make them more vulnerable to catching the virus again. However, generally speaking, Purba et al. and WHR [37,38] found that women suffered more from the pandemic compared to men. Similarly, the interaction effect (*β*_14_) of being older than 40 years of age and being vaccinated is found to be negative, which shows that younger people benefited relatively more than older people from vaccination in terms of their SWB. One possible reason could be that seniors or older population are usually at higher health risk and may need extra COVID-19 vaccine booster shots for higher health and wellbeing. However, this is in contrast to WHR [38], which shows the younger people suffered more during the pandemic. Evidence on European countries has shown that women and younger people have a lower acceptance rate of the COVID-19 vaccine [17] and for those of other diseases in the past. Age and sex are prominent risk factors associated with COVID-19 vaccine mandates. For instance, Bardosh, Krug and Jamrozik et al. [39] suggest an expected net harm from COVID-19 vaccine boosters among young adult age groups (under 30 years old). Last but not least, the interaction effect (*β*_17_) of being healthy and being vaccinated is found to be positive and the strongest among all the set of control variables in Model (1). In other words, COVID-19 vaccination moderates the relationshipbetween SWB and self-reported health status positively.

The specification error test for our baseline Model (1) is given in Table 5. The variable –hatsq is found to be statistically insignificant, while the variable –hat is statistically significant, which indicates that all the relevant and important variables have been included in Model (1). 

## 4. Conclusions

Using original survey data for urban Pakistan, this paper tries to shed some light on the life satisfaction of households during the COVID-19 health crisis. The evidence is, by nature, preliminary. Our happiness model estimated during the crisis extends the conventional scope by adding the impact of COVID-19 vaccination as a further determinant of self-perceived happiness. In general, our results suggest that apart from the various socio-demographic factors such as being younger (less than or equal to forty years of age) or healthier, more educated, richer, having children or living as married couple, COVID-19 vaccination is positively as well as significantly related to households’ happiness. However, the vaccination interaction effect of being male or being older (above forty years of age) is negative, indicating that the SWB of males or older populations was disproportionally or poorly affected as compared to their counterparts. In a developing country like Pakistan, men (who are mostly the bread winners) are usually at higher health risk of catching the virus again because of the nature of their job outside the home environment; older people, having weak immune systems, may need some extra COVID-19 vaccine booster shots for higher health and wellbeing. Lastly, the relationship between being healthy and subjective well-being was the strongest and positive considering the effect of vaccination. 

Our study is an early attempt in contributing to the analysis of happiness during the COVID-19 pandemic in this part of the developing world. The results established here turn out to be not disproportionally different across regions, although the overall socio-economic situation may differ in each region. Whilst effects might have varied across developing countries in magnitude, our analysis may also provide some important insights into the developing world in general. One important lesson may be that public support is particularly needed in those cases where public health education and services are weak. At the end of the day, the COVID-19 crisis marked a crisis for humanity requiring strong public, family and social support to avoid lasting scars on subjective life satisfaction.

## Figures and Tables

**Table 1 ijerph-20-06545-t001:** Descriptive statistics of happiness metric.

	Happiness (1–4)		
	Pooled	2021	2022
Mean	2.10	1.95	2.25
Standard deviation	1.10	1.11	1.09
Frequency of value:			
1	34.5%	39.5%	29.5%
2	35.5%	35.5%	35.5%
3	17.75%	16.5%	19%
4	12.25%	8.5%	16%

**Table 2 ijerph-20-06545-t002:** Descriptive statistics of socio-demographic variables.

	Mean/in Percent (%)		
	Pooled	2021	2022
Male	62%	64%	60%
Female	38%	36%	40%
Married couple	49%	52%	46%
Age	34 years	37 years	31 years
Education	12 years	12 years	12 years
Healthy	46%	42%	50%
Vaccinated	62%	56%	69%
Childless	42%	45%	39%
Unemployed	45%	47%	43%
Household’s monthly income in Pakistani rupees (PKR)	40,650	41,200	40,100
or	or		
in US dollars (USD)	264	268	260

**Table 3 ijerph-20-06545-t003:** Empirical results (ordered probit without vaccination effect and with vaccination effect).

	Without Vaccination Effect		With Vaccination Effect	
	Coef.	Robust Std. Err.	Coef.	Robust Std. Err.
Male_dummy_	−0.4900 ^*^	0.1976	−0.5400 ^**^	0.1970
Age	0.2987 ^*^	0.1413	0.2906 ^*^	0.1494
Age^2^	−0.0037 ^*^	0.0016	−0.0036 ^*^	0.0017
Years of education	0.0881 ^**^	0.0317	0.0881 ^**^	0.0317
Unemployed_dummy_	−0.6228 ^*^	0.3150	−0.6228 ^*^	0.3150
Log of monthly income	0.5677 ^*^	0.2351	0.5677 ^*^	0.2351
Childless_dummy_	−1.125 ^*^	0.4739	−1.125 ^*^	0.4739
Married couple_dummy_	1.3750 ^***^	0.3827	1.3750 ^***^	0.3827
(Married couple.Children)_dummy_	1.0952 ^**^	0.3764	1.0952 ^**^	0.3764
Healthy_dummy_	1.0833 ^**^	0.3294	1.5989 ^***^	0.1862
COVID-19 vaccinated_dummy_	-	*-*	1.0960 ^**^	0.3756
Region: Punjab_dummy_	1.4900	0.7906	1.4900	0.7906
Region: Sind_dummy_	0.9986	0.8165	0.9986	0.8165
Region: KPK_dummy_	0.9899	0.8660	0.9899	0.8660
Region: Baluchistan	Reference		Reference	
/cut 1	42.4900	20.0735	42.5201	20.1000
/cut 2	45.2811	21.3980	45.3840	21.4010
/cut 3	48.1372	22.1742	48.1573	22.2032
Log likelihood	−4.1362		−4.1454	
Observations	4500		4500	
LR *χ*^2^ (k − 1)	LR *χ*^2^ (12) = 39.79		LR *χ*^2^ (13) = 42.99	
Prob > *χ*^2^	0.0000		0.0000	
Pseudo *R*^2^	0.6549		0.6655	

Note. ^*^, ^**^ and ^***^ indicate 5%, 1% and 0.1% levels of statistical significance, respectively. *p* value ≤ 0.05, *p* value ≤ 0.01 and *p* value ≤ 0.001 were considered as the thresholds of the significance levels at 5%, 1% and 0.1%, respectively.

**Table 4 ijerph-20-06545-t004:** Ordered probit regressions with vaccination-interaction effects.

Vaccination-Interaction Effects	Separate Tests								Joint Test
Specification	(1)	(2)	(3)	(4)	(5)	(6)	(7)	(8)	(9)
Male_dummy_	**−0.5500** ^**^	**−0.5404** ^**^	**−0.5400** ^**^	**−0.5400** ^**^	**−0.5359** ^**^	**−0.5400** ^**^	**−0.5400** ^**^	**−0.5400** ^**^	**−0.4412** ^*^
	0.1971	0.1970	0.1970	0.1970	0.1899	0.1970	0.1970	0.1970	0.1880
Age	**0.2906** ^*^	**0.2899** ^*^	**0.2906** ^*^	**0.2906** ^*^	**0.2904** ^*^	**0.2906** ^*^	**0.2906** ^*^	**0.2906** ^*^	**0.2809** ^*^
	0.1494	0.1455	0.1494	0.1494	0.1491	0.1494	0.1494	0.1494	0.1405
Age^2^	**−0.0036** ^*^	**−0.0036** ^*^	**−0.0036** ^*^	**−0.0036** ^*^	**−0.0036** ^*^	**−0.0036** ^*^	**−0.0036** ^*^	**−0.0036** ^*^	**−0.0035** ^*^
	0.0017	0.0016	0.0017	0.0017	0.0017	0.0017	0.0017	0.0017	0.0015
Years of education	**0.0881** ^**^	**0.0881** ^**^	**0.0881** ^**^	**0.0881** ^**^	**0.0881** ^**^	**0.0881** ^**^	**0.0881** ^**^	**0.0881** ^**^	**0.0881** ^**^
	0.0317	0.0317	0.0317	0.0317	0.0317	0.0317	0.0317	0.0317	0.0309
Unemployed_dummy_	**−0.6228** ^*^	**−0.6228** ^*^	**−0.6228** ^*^	**−0.6228** ^*^	**−0.6228** ^*^	**−0.6228** ^*^	**−0.6228** ^*^	**−0.6228** ^*^	**−0.6228** ^*^
	0.3150	0.3150	0.3150	0.3150	0.3150	0.3150	0.3150	0.3150	0.3142
Log of monthly income	**0.5677** ^*^	**0.5677** ^*^	**0.5677** ^*^	**0.5677** ^*^	**0.5677** ^*^	**0.5677** ^*^	**0.5677** ^*^	**0.5677** ^*^	**0.5677** ^*^
	0.2351	0.2351	0.2351	0.2351	0.2351	0.2351	0.2351	0.2351	0.2345
Childless_dummy_	**−1.1250** ^*^	**−1.1250** ^*^	**−1.1250** ^*^	**−1.1250** ^*^	**−1.1250** ^*^	**−1.1250** ^*^	**−1.1250** ^*^	**−1.1250** ^*^	**−1.1250** ^*^
	0.4739	0.4739	0.4739	0.4739	0.4739	0.4739	0.4739	0.4739	0.4736
Married couple_dummy_ couple	**1.3750** ^***^	**1.3750** ^***^	**1.3750** ^***^	**1.3750** ^***^	**1.3750** ^***^	**1.3750** ^***^	**1.3750** ^***^	**1.3750** ^***^	**1.3750** ^***^
	0.3827	0.3827	0.3827	0.3827	0.3827	0.3827	0.3827	0.3827	0.3825
(Married couple. Children)_dummy_	**1.0952** ^**^	**1.0952** ^**^	**1.0952** ^**^	**1.0952** ^**^	**1.0952** ^**^	**1.0952** ^**^	**1.0952** ^**^	**1.0952** ^**^	**1.0952** ^**^
	0.3764	0.3764	0.3764	0.3764	0.3764	0.3764	0.3764	0.3764	0.3763
Healthy_dummy_	**1.5989** ^***^	**1.5979** ^***^	**1.5989** ^***^	**1.5989** ^***^	**1.4909** ^***^	**1.5989** ^***^	**1.5989** ^***^	**1.5989** ^***^	**1.5609** ^***^
	0.1862	0.1854	0.1862	0.1862	0.1780	0.1862	0.1862	0.1862	0.1801
COVID-19 vaccinated_dummy_	**0.9990** ^***^	**0.8960** ^***^	**1.0960** ^**^	**1.0960** ^**^	**1.0890** ^***^	**1.0960** ^**^	**1.0960** ^**^	**1.0960** ^**^	**1.0698** ^***^
	0.2690	0.2679	0.3756	0.3756	0.3049	0.3756	0.3756	0.3756	0.2989
Region: Punjab_dummy_	**1.4900**	**1.4900**	**1.4900**	**1.4900**	**1.4900**	**1.4900**	**1.4900**	**1.4900**	**1.4900**
	0.7906	0.7906	0.7906	0.7906	0.7906	0.7906	0.7906	0.7906	0.7906
Region: Sind_dummy_	**0.9986**	**0.9986**	**0.9986**	**0.9986**	**0.9986**	**0.9986**	**0.9986**	**0.9986**	**0.9986**
	0.8165	0.8165	0.8165	0.8165	0.8165	0.8165	0.8165	0.8165	0.8165
Region: KPK_dummy_	**0.9899**	**0.9899**	**0.9899**	**0.9899**	**0.9899**	**0.9899**	**0.9899**	**0.9899**	**0.9899**
	0.8660	0.8660	0.8660	0.8660	0.8660	0.8660	0.8660	0.8660	0.8660
Region: Baluchistan	**Reference**	**Reference**	**Reference**	**Reference**	**Reference**	**Reference**	**Reference**	**Reference**	**Reference**
(Male. vaccinated)_dummy_	**−0.7500** ^***^								**−0.7502** ^***^
	0.1531								0.1529
(Age. vaccinated)_dummy_		**−0.4953** ^**^							**−0.4956** ^**^
		0.1847							0.1844
(Income. vaccinated)_dummy_			**0.6666**						**0.6670**
			0.4505						0.4677
(Married couple. vaccinated)_dummy_				**1.5000**					**1.5010**
				1.0865					1.0875
(Healthy. vaccinated)_dummy_					**2.2222** ^**^				**2.2224** ^**^
					0.7027				0.7025
(Unemployed. vaccinated)_dummy_						**−0.5010**			**−0.5015**
						0.8539			0.8544
(Childless. vaccinated)_dummy_							**−0.8333**		**−0.8336**
							0.5743i		0.5746
(Education. vaccinated)_dummy_								**0.1428**	**0.1430**
								0.2259	0.2261
/cut 1	**42.5502**	**42.5504**	**42.5505**	**42.5501**	**42.5508**	**42.5509**	**42.5501**	**42.5506**	**42.7505**
	20.1325	20.1307	20.1308	20.1301	20.1309	20.1308	20.1300	20.13041	20.34222
/cut 2	**45.4830**	**45.4832**	**45.4833**	**45.4831**	**45.4838**	**45.4839**	**45.4829**	**45.4836**	**46.1812**
	21.4112	21.4116	21.4117	21.4115	21.4121	21.4122	21.4112	21.4119	21.48230
/cut 3	**48.1750**	**48.1754**	**48.1755**	**48.1751**	**48.1758**	**48.1759**	**48.1750**	**48.1756**	**48.3025**
	22.2522	22.2528	22.2529	22.2521	22.2530	22.2531	22.2521	22.2528	22.4650
Number of observations	4500	4500	4500	4500	4500	4500	4500	4500	4500
LR *χ*^2^(k − 1)	LR *χ*^2^(14) = 45.92	LR *χ*^2^(14) = 45.94	LR *χ*^2^(14) = 45.95	LR *χ*^2^(14) = 45.91	LR *χ*^2^(14) = 45.98	LR *χ*^2^(14) = 45.99	LR *χ*^2^(14) = 45.90	LR *χ*^2^(14) = 45.96	LR *χ*^2^(21) = 69.15
Prob *> χ*^2^	0.0000	0.0000	0.0000	0.0000	0.0000	0.0000	0.0000	0.0000	0.0000
Pseudo *R*^2^	0.6705	0.6706	0.6707	0.6704	0.6708	0.6709	0.6700	0.6706	0.7400
Log likelihood	−4.1555	−4.1556	−4.1557	−4.1554	−4.1558	−4.1559	−4.1550	−4.1556	−4.2250

Note: Regression coefficients are in bold and standard errors appear below them. ^*^, ^**^ and ^***^ indicate 5%, 1% and 0.1% levels of statistical significance, respectively. *p* value ≤ 0.05, *p* value ≤ 0.01 and *p* value ≤ 0.001 were considered as the thresholds of the significance levels at 5%, 1% and 0.1%, respectively.

**Table 5 ijerph-20-06545-t005:** Specification error test: baseline Model (1).

**Specification Error Test**		
**Number of Obs**	=	**4500**
LR *χ*^2^ (2)	=	66.03
Prob *> χ*^2^	=	0.0000
Pseudo *R*^2^	=	0.6705
Log likelihood	=	−4.3223
Dependent variable: SWB		
Independent	coef.	Robust
Variable		Std. Err.
-hat	1.2351 ^*^ [0.040]	0.6008
-hatsq	−0.0254 [0.587]	0.0467
/cut1	42.5472	19.9802
/cut2	45.3125	20.7875
/cut3	48.2312	21.9520

Note. ^*^ indicate 5% levels of statistical significance, respectively; *p* values are given in square brackets. *p* value ≤ 0.05, *p* value ≤ 0.01 and *p* value ≤ 0.001 were considered as the thresholds of the significance levels at 5%, 1% and 0.1%, respectively.

## Data Availability

The data sets used in the study are not publicly available due to privacy and ethical concerns. However, all the descriptive data are available within the paper.

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
