# Peer review of "Subjective Well-Being, Health and Socio-Demographic Factors Related to COVID-19 Vaccination: A Repeated Cross-Sectional Sample Survey Study from 2021–2022 in Urban Pakistan"

_ijerph, 2023, doi:10.3390/ijerph20166545_

Round 1

Reviewer 1 Report

Journal: International Journal of Environmental Research and Public Health

Title: Subjective well-being, health and socio-demographic factors related to COVID-19 vaccination: a repeated cross-sectional sample survey study, 2021-2022 in urban Pakistan

Comments: The vaccination is important to contain the spread of COVID-19. This study investigated the effects on the intention in urban Pakistan by health, socio-demographic factors, and subjective well-being. About 4500 households were involved through questionnaires from household’s head. An ordered probit regression model was used to estimate the relationships between the factors and COVID-19 vaccine. The results indicated coronavirus vaccination is the fourth factor for subjective well-being followed after being healthy, educated, and richer. The relationship between being a male and subjective well-being was negative considering the effect of vaccination. The narrations of this article are explicit, and the results are valid. There are some comments expected to improve the quality of this study:

(1) In the introduction section, the relationship between happiness and novel corona virus was weak, especially the vaccination. Please add some descriptions to emphasize the relevance.

(2) In section 2.1, were all the questionnaires valid, or there was the process of dada cleansing?

(3) The definitions for the variables, such as health status, material status, and vaccination status in model (1) were equivocal, and please clear them.

(4) In section 3.1, the relationships between gender, age, and being vaccinated against the virus were contrast with previous reports, but the reasons were not showed. Please add it.

The English language in this article is generally good, and  some minor mistakes, such as articles and plurals, should be checked.

Reviewer 2 Report

Thank you very much for the opportunity to review this article about Subjective well-being, health, and socio-demographic factors related to COVID-19 vaccination: a repeated cross-sectional sample survey study, 2021-2022 in urban Pakistan. Overall I found it to be a bold article that offers a different approach and, therefore, may offer a valuable perspective.

However, some minor aspects are amenable to revision. These are as follows:

In the Methods section, I propose that at the beginning, they describe in a general way the type of study being performed. Then, the eligible population and its characteristics. The latter, together with the selection criteria, is important to be well described since one of the main problems of this type of study lies precisely in the potential for selection bias.

Although specific characteristics of the sample, such as size, are discussed, evaluating the potential selection bias would be important.

Review self-citation references:

16.   Shams K. Determinants of Subjective Well-being and Poverty in Rural Pakistan: A Micro-Level Study. Social Indicators Research. 2014; 119: 1755-1773.

17.   Shams K,  Kadow A. Happiness across the life span: Evidence from urban Pakistan. FWU Journal of Social Sciences. 2018; 12(1): 17-30.

18.   Shams K. Developments in the Measurement of Subjective Well-being and Poverty: An Economic Perspective. Journal of Happiness Studies. 2016; 17(6): 2213-2236.

Reviewer 3 Report

In the manuscript, the authors examined Subjective well-being, health and socio-demographic factors related to COVID-19 vaccination: a repeated cross-sectional sample survey study, 2021-2022 in urban Pakistan.

However, there are some remarks.

1. Line 18-27. The abstract should specify the results obtained.

2. Line 42-52. Show the socio-economic profile of the population in the context of this study. Show the number (%) of men, % of older people, etc.

3. At the end of the Introduction, write the purpose of this study.

4. In the Materials and Research Methods section, describe in more detail the calculation apparatus and the principle of conducting research. Namely, how the effect of vaccination was assessed, methods of statistical data processing, etc.

5. The level of statistical significance was calculated. By what method? Page 6.

6. Table 4. Explanation of model numbering is required.

7. Redo the conclusion. Add specific research findings based on the findings.

These comments are indicated in the manuscript. The file is attached to the letter!

The text of the manuscript is written in good English. However, there are minor errors and inaccuracies. Therefore, a small editorial change is required.

Round 2

Reviewer 1 Report

The authors have made explicit responses and corrections, and it's recommended to be accepted for publication at present version.